# Development of Zika Virus E Variants for Pseudotyping Retroviral Vectors Targeting Glioblastoma Cells

**DOI:** 10.3390/ijms241914487

**Published:** 2023-09-23

**Authors:** Vivien Grunwald, Hai Dang Ngo, Jan Patrick Formanski, Jana Sue Jonas, Celine Pöhlking, Birco Schwalbe, Michael Schreiber

**Affiliations:** 1Department of Virology, LG Schreiber, Bernhard Nocht Institute for Tropical Medicine, 20359 Hamburg, Germanyjan.patrick.formanski@studium.uni-hamburg.de (J.P.F.); janasue.jonas@haw-hamburg.de (J.S.J.); celine.poehlking@haw-hamburg.de (C.P.); 2Department of Neurosurgery, Asklepios Kliniken Hamburg GmbH, Asklepios Klinik Nord, Standort Heidberg, 22417 Hamburg, Germany; b.schwalbe@asklepios.com

**Keywords:** glioblastoma, oncolytic viruses, pseudotypes, Zika virus, retroviral vector, flavivirus pseudotypes, HIV-1, transmembrane domain, gp41, prME

## Abstract

A fundamental idea for targeting glioblastoma cells is to exploit the neurotropic properties of Zika virus (ZIKV) through its two outer envelope proteins, prM and E. This study aimed to develop envelope glycoproteins for pseudotyping retroviral vectors that can be used for efficient tumor cell infection. Firstly, the retroviral vector pNL*luc*AM was packaged using wild-type ZIKV E to generate an E-HIV*luc* pseudotype. E-HIV*luc* infection rates for tumor cells were higher than those of normal prME pseudotyped particles and the traditionally used vesicular stomatitis virus G (VSV-G) pseudotypes, indicating that protein E alone was sufficient for the formation of infectious pseudotyped particles. Secondly, two envelope chimeras, E41.1 and E41.2, with the E wild-type transmembrane domain replaced by the gp41 transmembrane and cytoplasmic domains, were constructed; pNL*luc*AM or pNL*gfp*AM packaged with E41.1 or E41.2 constructs showed infectivity for tumor cells, with the highest rates observed for E41.2. This envelope construct can be used not only as a tool to further develop oncolytic pseudotyped viruses for therapy, but also as a new research tool to study changes in tumor cells after the transfer of genes that might have therapeutic potential.

## 1. Introduction

Glioblastoma, previously known as glioblastoma multiforme (GBM), is an extremely aggressive and difficult-to-treat form of brain tumor for which there is no cure [1]. Treatment options for glioblastoma depend on the location and size of the tumor, as well as the age and overall health of the patient. It should be noted that treatment for glioblastoma is often palliative, meaning that it is aimed at relieving symptoms and improving quality of life, rather than curing the cancer [2].

The most common treatment options for glioblastoma are (i) surgery, which is often the first step in treating glioblastoma, as it can help to remove as much of the tumor as possible; (ii) radiation therapy, which is used after surgery to destroy any remaining cancer cells; and (iii) chemotherapy to kill cancer cells with certain drugs [3].

Today, glioblastoma multiforme (GBM) remains a difficult and aggressive form of brain cancer with very limited treatment options. Research into new therapies and treatment approaches is currently being pursued at all levels [4].

However, in addition to therapeutic approaches that directly target tumor cells, the tumor microenvironment also plays a role in supporting tumor growth. Tumor-associated microglia/macrophages and tumor-infiltrating CD4+ and CD8+ lymphocytes also play important roles in tumor progression and overall survival. Activation of Axl by macrophages, one of the ZIKV receptors, and the support of ZIKV oncolytic activity by CD8+ lymphocytes are important aspects for the development of GBM therapies based on ZIKV properties [5,6,7].

A promising new method for treating cancer is oncolytic tumor therapy, which uses viruses to target cancer cells. The underlying idea is to use viruses that naturally attack and destroy cells through various mechanisms, such as apoptosis, pyroptosis, and necroptosis. While necroptosis and pyroptosis lead to the release of immunostimulatory molecules and other pro-inflammatory signals into the cellular environment, apoptosis is considered “silent” and thereby attenuates subsequent immune responses [8,9]. There are several types of viruses that have been tested in glioblastoma trials—for example, parvoviruses, adenoviruses, herpes simplex, polio, and reoviruses [10]. Despite their safety and efficacy, these viruses have shown in animal and preclinical studies that their efficacy is still insufficient. Notably, they have shown little or no efficacy in subsequent clinical trials [11].

Overall, the natural mechanisms by which viruses destroy cells are complex, vary depending on the virus and cell type, and in most cases are also completely unknown. In contrast, viral entry and viral receptors, i.e., the tropism of a virus, are well studied.

Neurotropism refers to the ability of viruses to infect and replicate in cells of the nervous system. Examples of neurotropic viruses include herpes simplex virus, which can cause herpes encephalitis and other neurological disorders, and the rabies virus, which causes fatal encephalitis in humans and animals [12,13]. Other viruses that can exhibit neurotropism include poliovirus [14,15], West Nile virus [16], and especially Zika virus (ZIKV), which causes microcephaly in neonates [17].

ZIKV has a remarkable tropism for neural stem/progenitor cells (NSCs/NPCs), astrocytes, oligodendrocyte progenitor cells, and microglial cells [18,19]. In addition, ZIKV preferentially infects and kills glioblastoma stem cells (GSCs), which share similarities with normal NSCs/NPCs, including intrinsic resistance to radiotherapy and chemotherapy [20,21,22]. GSC-specific tropism results from the expression of Sox2 and integrin αvβ5 in GBM stem cells [23]. The effect against GSCs is not a general property of neurotropic flaviviruses, as West Nile virus indiscriminately kills both tumoral and normal neurons [16,24,25]. However, ZIKV strongly decimates patient derived GSCs grown in culture and in organoids [26].

A ZIKV that was engineered to selectively target and infect glioblastoma cells showed significant tumor cell death in a GSC-derived orthotopic model, in which human glioma stem cells were implanted into the brains of immunocompromised mice [27]. Therefore, one approach is to exploit the neurotropic properties of ZIKV, either in the form of an attenuated, fully live virus [27] or in the form of the outer envelope proteins as part of a pseudotyped viral particle [28,29]. Improving targeted gene transfer, especially by pseudotyping retroviral vectors with viral glycoproteins that promote high infection rates, is of utmost importance in cancer research and gene transfer research in general. Altogether, the ZIKV envelope proteins are excellent candidates to generate pseudotyped virus particles with neurotropic properties for cell-directed gene transfer.

One hurdle to overcome is that Zika virus, like many other RNA viruses, uses the endoplasmic reticulum (ER) membrane as a site for viral protein synthesis and replication. At the ER, viral RNA is translated into a polyprotein that is processed into individual viral proteins by host and viral proteases. The ER membrane also provides a platform for the assembly and formation of ZIKV particles, with the two proteins prM and E as its outer envelope. The E protein is responsible for binding to receptors on the surface of host cells and initiates the process of membrane fusion to allow viral entry [30].

Pseudotyped viruses are viral particles that contain the envelope proteins of one virus but the core genetic material of another virus. For example, a pseudotyped virus may contain the envelope proteins prM and E of ZIKV but the core and genetic material of human immunodeficiency virus type-1 (HIV-1). Initial reports indicate that such pseudotypes can not only be produced but are also capable of infecting cells isolated from GBM tumors [28,29]. However, the yield of pseudotypes generated in these reports is very low. The reason for the low pseudotype efficiency might be the different sites of virus formation. As already mentioned, ZIKV is formed at the ER membrane, while HIV-1, on the other hand, is formed at the outer cell membrane [31,32]. Therefore, it is an important step to modify the structure of the ZIKV envelope proteins in such a way that they will be transported to the outer membrane, following the path of HIV-1 assembly.

In a first step, we examined HIV-1 viral particles pseudotyped with ZIKV E protein and compared them with prME pseudotypes to show that the E protein generates infectious pseudotypes in the absence of the prM protein. In a second step, we generated pseudotypes in which the transmembrane domain (TM) of E was exchanged against the TM and cytoplasmic (CY) domains of the HIV-1 glycoprotein gp41. Finally, E and E^∆TM^-gp41^TMCY^ pseudotypes were tested for infection of U87 cells and freshly isolated cells from GBM tumor samples. The aim was to develop ZIKV E variants that show better infectivity compared to the known VSV-G pseudotype. These could be used in the future to introduce therapeutic genes into GBM tumor cells in a targeted manner.

## 2. Results

### 2.1. Immunostaining of Cells Isolated from GBM Tumors

Expression of the embryonic stem cell markers nanog, nestin, Oct4, and Sox2 is critical for the progression of various human malignancies, including brain tumors. Therefore, we used these markers to characterize cells isolated from GBM tumors, termed AKH-14 and AKH-16. For immunostaining, AKH cell cultures were seeded into 96-well plates and grown in 50% hCSF, 45% DMEM, and 5% FBS. In addition, the cells were tested for the presence of Axl and integrin αvβ5, the two proposed ZIKV receptor molecules.

The first apparent difference was that astrocyte-like cells with long and highly branched filopodia grew alongside other glioma cells in both AKH cultures. These filopodia-rich cells could be distinguished from other cells present in the cultures by phalloidin staining alone, which was particularly evident in AKH-14/Sox2 and AKH-16/Axl. For example, the filopodia-rich cell in the center of the AKH-16/Axl image was actin-negative and surrounded by many actin-positive cells, forming a layer on which the central cell was located, with its spreading radius of approximately 200 µm. Such an effect of astrocyte–cell interaction was not seen in the AKH-14 cultures. However, as shown in Figure 1, all cells except for filopodia-rich cells showed positive actin expression (phalloidin staining).

Axl, also known as AXL tyrosine kinase, is a cell surface receptor protein and belongs to the TAM receptor tyrosine kinase family, which also includes TYRO3 and MER. Axl plays roles in cell growth, survival, adhesion, and migration, and is controversially discussed as a potential Zika virus receptor [33,34]. Interestingly, tumor-associated microglia/macrophages, which constitute an essential part of the GBM microenvironment, activate Axl expression and produce tumor-supporting growth factors [7,35]. As shown in Figure 1, Axl expression was observed in all AKH cells, in both phalloidin-negative and phalloidin-positive cells.

Integrin αvβ5 has attracted attention due to its involvement in tumor growth, invasion, and metastasis. It promotes cancer cells’ survival, proliferation, and migration by mediating interactions between cancer cells and the surrounding microenvironment [36]. Like Axl, integrin αvβ5 has been shown to be a receptor candidate for ZIKV, and its importance for viral infection was demonstrated in glioma cells [37,38]. AKH-14 cells were partially positive for integrin αvβ5, especially in round-shaped cells. AKH-16 cells were all positive for integrin αvβ5, which was predominantly localized in the nucleus compared with Axl.

Nanog is a transcription factor that is well known for its role in maintaining pluripotency and self-renewal in embryonic stem cells. In glioblastoma, nanog expression has been reported in a subset of tumor cells that are thought to contribute to tumor initiation, progression, and therapy resistance [39]. In AKH-14 cells, nanog was detected in cells of a more round-shaped morphology (yellow arrow). These cells were positive for actin but did not show the typical actin-fiber structures of the larger glioma cells. In AKH-16, nanog was detected in cells with short filopodia that were also actin-positive. In contrast to the surrounding cells, these cells were in the late phase of division, as two nuclei appeared in the DAPI staining. In contrast to the AKH-14 cultures, nanog-positive cells were observed rather rarely in AKH-16 and also had a different morphology compared to the round-shaped nanog-positive cells identified in AKH-14 cultures.

Nestin is often expressed in GBM tumor cells [40]. In AKH-14 cells, nestin was clearly detected in filopodia-rich astrocytes and in the form of small granules in cells with the roundish morphology characteristic of nanog-positive cells (yellow arrow). In AKH-16, nestin positivity was seen in actin-negative cells. These cells show the morphology of a subset of primary tumor cells, as described by Veselska et al. [41].

While Oct4 is primarily associated with embryonic stem cells, studies have reported the expression of Oct4 in a subset of glioblastoma cells [42]. In both cell cultures, Oct4 was expressed in all types of cells. In particular, filopodia-rich astrocytes showed high Oct4 expression compared to actin-rich glioma cells. In AKH-14 cultures, Oct4 expression was also detected in actin-positive cells with a roundish morphology (yellow arrow).

Sox2 promotes the proliferation, survival, and invasion of glioblastoma cells. It regulates genes involved in cell-cycle regulation, DNA repair, and cell adhesion, and is also considered to be an upregulator of integrin αvβ5 expression [23,42]. In AKH-14 cultures, Sox2 expression was clearly associated with the filopodia-rich astrocyte phenotype, and the actin-rich glioma cells were Sox2-negative. In AKH-16 cultures, Sox2 expression was detected in all cell types, and no difference was observed between actin-positive and actin-negative cells. As described earlier, the standard U87 cells were positive for Axl, integrin αvβ5 [29], Sox2, Oct4, and nestin, while they were negative for nanog.

In summary, in AKH-14 cultures, filopodia-rich astrocytes tested positive for Axl, integrin αvβ5, nestin, Sox2, and Oct4 expression. Large actin-rich glioma cells tested positive for Axl, integrin αvβ5, and Oct4. Actin-positive cells with a roundish morphology were found to be positive for nanog, nestin, and Oct4. In AKH-16 cultures, astrocyte-like cells were positive for Axl, integrin αvβ5, Sox2, and Oct4. Large actin-rich glioma cells were positive for Axl, integrin αvβ5, Sox2, and Oct4. In contrast to AKH-14, we could not detect cells with a roundish morphology in the AKH-16 cultures, but cells that were actin-negative and positive for nestin were detected.

### 2.2. Infection of Cells with HIVluc Pseudotypes with ZIKV Wild-Type TM^E^-Containing Envelope

As previously shown, the ZIKV Z1-HIV*luc* (prME) and ∆pr-HIV*luc* (ME) pseudotypes are functional in infecting glioma cell lines and isolated GBM tumor cells [29]. Referring to Figure 2, it can be seen that the E protein alone, without the presence of prM or M proteins, forms a functional envelope that enables pseudotype infection of the U87 glioma cell line and cells isolated from GBM tumors, termed AKH-14 and AKH-16, respectively.

Z1-HIV*luc* infection of U87 cells was low compared to E1- and E2-HIV*luc* infection. Interestingly, both E pseudotypes showed infection rates above the G-HIV*luc* level. When AKH-14 was infected, the E1- and E2-HIV*luc* infection rates were significantly higher than those of Z1-HIV*luc* and G-HIV*luc*, both used for comparison. The data demonstrate that the ZIKV E protein alone, including its normal transmembrane domain TM(E) (Figure A2), can form a functional pseudotype shell. The infection rates for E1-HIV*luc* and E2-HIV*luc* were equal to or even higher than the rates observed with the traditional VSV-G pseudotype used to infect GBM and other cell types in general.

### 2.3. Infection of Cells with E^ΔTM^gp41^TMCY^-HIVluc Pseudotypes

Next, we examined pseudotypes with an altered ZIKV E transmembrane domain. For the expression of E41.1 and E41.2, we used GCCGCCGCC as a Kozak sequence, as we did previously for Z1, E1, and E2. The E transmembrane domain (TM) was omitted, and E^∆TM^, ending with the amino acid sequence AFKS, was fused to the TM and cytoplasmic domain (CY) of gp41, starting with LFIMI. In AKH cells, the levels of E41.1-HIV*luc* and E41.2-HIV*luc* pseudotype infection were also higher compared with G-HIV*luc* and comparable with the results of E1-HIV*luc* and E2-HIV*luc* infection, as shown in Figure 2B. In addition, ZIKV E variants with mixed ED12 and ED3 domains from Japanese encephalitis virus (JEV) or Usutu virus (USUV) were tested for comparison. None of these variants showed higher infection rates compared to the G-HIV*luc* control. This demonstrates that the ZIKV E^∆TM^-gp41^TMCY^ fusion can create functional HIV-1 pseudotypes with high infectivity for AKH and U87 cells. This high infectivity was not enhanced by using ED3 or ED1/2 domains from neurotropic JEV or USUV, as shown in Figure 3.

### 2.4. Infection of AKH Cells with E41.1- and E41.2-HIVgfp Pseudotypes

AKH cells were infected with the appropriate HIV*gfp* pseudotypes to compare the luciferase concentrations in lysates of infected cells (Figure 3B) with the number of *gfp* signals from infected living cells. Therefore, AKH cells were infected with E41.1-, E41.2-, and G-HIV*gfp*. Infected cells were monitored by their green fluorescence on day three. As Figure 4 shows, the infection rates were comparable to the rates of the corresponding HIV*luc* pseudotypes, as shown in Figure 3B, with E41.2-HIV*gfp* showing the highest rate and G-HIV*gfp* showing the lowest. Thus, the *gfp* positivity of single live cells also demonstrated the efficiency of ZIKV E41 pseudotypes, with infection rates higher than those of the traditionally used VSV-G pseudotypes. This can be seen as further evidence for the better performance of the new generation of ZIKV/HIV pseudotypes, as the latter performed as well as or better than G-HIV*luc* or G-HIV*gfp*.

## 3. Discussion

Pseudotyped viruses are modified viruses in which the original surface proteins have been replaced by other proteins from a different virus or source. Through this process, the pseudotyped virus acquires the ability to invade the desired target cells. In the present case, these were GBM grade 4 tumor cells. Pseudotyping is widely used in research and gene therapy to study viral entry mechanisms, test antiviral drugs and, most importantly, develop safer viral vectors for gene delivery. Pseudotyping can also be used to engineer and individualize viral envelope proteins to efficiently mediate viral entry. Efficient gene transfer requires such a special envelope protein, and its improvement is of utmost importance. Since the retroviral vectors pNL*luc*AM and pNL*gfp*AM do not carry a functional envelope or other packaging signals, they transfer their viral genome in a single round of infection, making them ideal tools for studying envelope-dependent entry.

A VSV-G pantropic pseudotype (here G-HIV*luc* or G-HIV*gfp*) is a commonly used standard in virology and molecular biology research [43]. We used them to compare infection results and clearly demonstrated that the new ZIKV pseudotypes were more suitable for the transduction of GBM cells. They are not just new candidates to further develop oncolytic pseudotype viruses; they are also a new and reliable tool to study the effects of therapeutic genes when transferred by ZIKV/HIV pseudotypes into cancer cells.

HIV-1, as part of the retrovirus family, is not usually considered to be an oncolytic virus, but its genome offers a wide range of possibilities for therapeutic gene transfer, including CRISPR/Cas9 and siRNA applications. HIV-1 was also one of the first viruses for which pseudotyping could easily be performed. However, unlike other viral candidates, the three HIV enzymes show high sensitivity to a variety of antiviral drugs. Thus, if HIV-1 is used as an oncolytic virus, it can be inhibited efficiently with drugs. As mentioned earlier, the production of HIV pseudotypes is straightforward as long as the foreign envelope protein in the pseudotype-producing cells follows the same production pathway. As the virus particles assemble and mature, they are enveloped by host cell membranes, into which the envelope proteins are incorporated by their TM domains. The nature of these TM domains decides the type of membrane used for viral assembly. HIV-1 uses the outer membrane, whereas ZIKV assembles at the ER membrane. Therefore, pseudotyping of HIV-1 by prME envelope proteins of ZIKV or other flaviviruses seems to be theoretically impossible. We have developed a method to produce ZIKV/HIV pseudotypes and demonstrated infection of glioma cell lines, opening the door to a new tool for GBM-targeted virotherapy. Using four different ZIKV/HIV pseudotypes, the Z1 prME envelope was selected, demonstrating that it can be used to transmit retroviral vectors into glioma cell lines [28]. This is the first report to show not only that a flavivirus-pseudotyped HIV-1 can be produced, but also that such pseudotypes bearing a ZIKV-derived prME envelope are capable of infecting GBM cell lines. In another study, we showed that the ZIKV prME-HIV pseudotype can also infect primary tumor cells isolated from grade 4 GBM tissue samples. In addition, the ZIKV envelope protein was successfully modified to match the ME structure of the matured, furin-protease-activated virus particle [29]. In all of these studies, the ZIKV envelope—prME and ME—was expressed exactly as it occurs in the original genome, with the M and E transmembrane regions controlling ER localization. This is suggestive that the efficiency of pseudotype formation and infection of the cells were low and far below the infection rates observed with the VSV-G-pseudotyped control.

In the present study, we considered three aspects that are important for the future development of a ZIKV/HIV pseudotype: the tumor cells used to study pseudotype infection, the use of only ZIKV E as part of a retroviral coat, and modifications to the E transmembrane domain. In previous studies [28,29], we isolated cells from normal tumor tissue, and since all of these cells showed the same morphology as adherent epithelial cells such as U87, U138, and U343 cells, we used them for infection studies, particularly because they were all positive for the ZIKV receptors Axl and integrin αvβ5. However, gliomas of grades 3 and 4 do not show a clear boundary with the healthy brain, as their growth is characterized by vascular and axonal migration of glioma cells with diffuse infiltration of surrounding tissue. It has been shown that a heterogeneous set of tumor cells, including GSCs, settle in specific areas of the tumor [44,45,46]. Therefore, we changed our strategy and collected GBM tissue samples localized adjacent to the subventricular zone (SVZ). The SVZ, a region along the lateral ventricle, contains several cell populations, including astrocyte-like neural stem cells (NSCs), which are thought to be cells of origin for GBM development [44,46,47]. There is also an association between highly tumorigenic cell subsets in brain tumors and the expression of the markers nanog, nestin, Oct4, and Sox2 [48,49]. Therefore, we examined cell cultures from tumor tissue adjacent to the SVZ (AKH-14 and AKH-16) for expression of these four markers. This brought the advantage that we no longer isolated a homogeneous cell population lacking astrocytes [29], but instead obtained a wide spectrum of very different cells from the tumor sample. The disadvantage is that, in cell cultures, the composition of the respective cell populations can change over time. Therefore, we use the freshly isolated AKH cell cultures immediately for all investigations in a timeframe of <2 months. In conclusion, the SVZ-proximal two-dimensional cell cultures (AKH-14, AKH-16) exhibited marked heterogeneity that probably better reflects the in vivo situation of tumor composition. This may have been due in part to the use of a medium containing 50% hCSF. This created a more physiological, brain-like environment that is known to promote astrocyte development, the formation of synapses, and the preservation of nestin^+^ and Sox2^+^ cells [50]. The use of hCSF and SVZ tumor tissue resulted in cell cultures with higher permissivity for infection. This was particularly evident for E1 and E2 pseudotypes. In contrast to the U87 cell line, we observed a significantly higher infection rate in the GBM cell isolates. Especially in the AKH-14 culture, the pseudotypes performed much better than the VSV-G control. One explanation could be that in these constructs the transmembrane domain of prM is not present. Thus, in contrast to our previous studies using a prME and ME envelope [29], the transmembrane domain of prM might be a relevant factor that directs it to and holds the protein in the ER membrane. Without prM (TM), the E protein can probably travel to the outer cell membrane more efficiently.

Unfortunately, there are a very limited number of publications to compare our pseudotype infection data with the existing literature to assess the extent to which the results obtained are consistent with or differ from previously published works [51].

There are two publications in which the ZIKV envelope was expressed on the surface of virus particles to produce vaccine candidates, using measles or baculovirus [52,53]. In the case of measles virus, the ZIKV prME was inserted into the viral genome without further modifications, like we did for the Z1-HIVluc pseudotype [14]. For expression on the surface of baculoviruses, the E protein was inserted between the signal and transmembrane regions of the gp64 envelope of *Autographa californica* multiple nuclear polyhedrosis virus (AcMNPV) [52]. These studies have shown that the ZIKV envelope can be expressed on the surface of virus particles. Most remarkably, this is also possible when the signals for membrane localization are exchanged. One study is following this idea [51]. For this purpose, the ZIKV-E protein was tagged with the appropriate localization signals of VSV to generate pseudotyped particles. The TM and CY domains of the VSV-G envelope were used to express an E/G-TC fusion under the control of the CMV promoter in the original VSV-G expression vector pMD2.G. In conjunction with a lentiviral vector (pLV-eGFP) [54] and a packaging plasmid (psPAX2), the co-responsive pseudotype was shown to efficiently infect renal cells [51]. Using this approach, E/G-TC pseudotypes showed levels of infectivity above those of the VSV-G pseudotype. These results are consistent with and confirm our early data showing that retroviral vectors can be packaged with a ZIKV envelope [28]. Using this approach, the E/G-TC pseudotypes exhibited higher infectivity than the VSV-G pseudotype. Taken together, these data can be interpreted as indicating that two factors are critical for infection studies with ZIKV/HIV pseudotypes: first, it seems important to improve the efficiency of ZIKV E by optimizing the transmembrane and signaling sequences, and second, the sensitivity of cells to ZIKV, along with their E-related permissiveness, plays an important role for further improvements of such pseudotypes. Since we were working with retroviral vectors that produce the HIV-1 core, we used gp41 as a template to replace the ZIKV E TM domain. In general, gp41 contains three domains: the ectodomain, the TM domain, and the CY domain. The CY domain is important for linking the membrane-anchored protein to HIV-1 *gag* and *gagpol* precursors. Even more important is the *gag*-related p6-domain-mediated activation of the ESCRT machinery, which causes budding of the viral particle. For these reasons, we constructed ZIKV E variants that carry the TM and CY domains of the gp41. Compared with the infection rates of E1- and E2-HIV*luc*, no significant increase in infectivity was observed using E41.1- or E41.2-HIV*luc*. However, with respect to E41 constructs, a significant increase in pseudotype infection of U87 cells was observed. Infection experiments with HIV*gfp* pseudotypes also supported the viability of the ZIKV/HIV pseudotypes and showed again that their infection rates were higher compared to those of the G-HIV*gfp* control. In addition to G-HIV*luc* used for comparison, we constructed four E41 pseudotypes based on the USUV and JEV E domains. Because both viruses are highly pathogenic and have neurotoxic potential [55,56], we tested mixed versions of the ZIKV/JEV and ZIKV/USUV E proteins as pseudotype envelopes. All four pseudotypes were infectious, but their infection rates were significantly lower than those of G-HIV*luc*. Therefore, no increase in infectivity by adding USUV or JEV ED3 or ED12 domains to ZIKV E was observed. In our opinion, ZIKV E is the most promising envelope to date for the development of an oncolytic pseudotype virus against GBM.

One of the most promising strategies for cancer treatment is targeted therapy, a form of treatment that targets specific proteins or genes in cancer cells [57]. The big question, however, is how therapeutic genes can be transported with high efficiency to their target site within the cancer cell. Viruses are excellent vehicles for introducing genetic material into cells. Because they can normally infect a very broad range of cells, they are not truly targeted transporters for therapeutic genes. The pseudotype approach is a very useful technique for designing and testing viral envelopes specific for certain surface markers. In combination with retroviral vectors, this offers a new and promising opportunity for the development of oncolytic pseudotyped viruses. Furthermore, the triggered antitumor effects of potential therapeutic genes can be studied when introduced into cancer cells with these new tools.

## 4. Materials and Methods

### 4.1. Plasmids

The plasmid pME is a modified version of pcDNA3.1 lacking the PvuII-PvuII fragment; pME was used for ZIKV prME expression (pME-Z1), and pME-JE (Japanese encephalitis virus (JEV), GenBank HE861351), pME-D2 (Dengue 2 (DENV2), (GenBank U87411), and pME-USU (Usutu virus (USUV), GenBank HE599647) were generated by the same procedure [28]. For making E-gp41 constructs, an HIV-1 gp160 expression vector—pSVATG*rev*, a gift from H. Schaal (Heinrich-Heine University, Düsseldorf, Germany)—was used [58]. pNL*luc*AM and pNL*gfp*AM, lentiviral vectors with the firefly luciferase gene (*luc*) or green fluorescence gene (*gfp*), were gifts from A. Trkola and N. Friedrich (Institute for Medical Virology, University Zürich, Switzerland) [59]. Detailed maps of the pSVATG*rev* cloning site and the lentiviral vectors are shown in Figure A1. The plasmid pCMV-VSV-G, expressing the envelope G from vesicular stomatitis virus (VSV), was from B. Weinberg, distributed by Addgene (#8454, Teddington, UK).

### 4.2. Cells

COS-1 cells (CVCL_0223) were provided by the Friedrich Löffler Institute, Riems-Greifswald, Germany. U87 cells from Dr. Bruce Chesebro were obtained through the NIH AIDS Reagent Program, Division of AIDS, NIAID, NIH [60]. Cells from grade 4 GBM tumor tissue samples were isolated as previously described in detail [29]. Briefly, tissue samples were squeezed through a sterile cell strainer (70 µm mesh size; Fisherbrand, Schwerte, Germany) into 50 mL tubes, and the cell suspension was washed with DMEM/10% FBS. Cells were suspended in medium (50% hCSF, 45% DMEM, and 5% FBS) and placed in cell culture flasks (Cell+™, Sarstedt, Nümbrecht, Germany). Cell growth was monitored daily for the appearance of adherent cells. Non-adherent cells were washed away when adherent cells reached a density of 40–50%. Tumor operations were performed at the Asklepios Klinik Nord-Heidberg (Hamburg, Germany). The study design, isolation of GBM cells for pseudotype infection, and the use of hCSF were approved by the Ethical Commission of the Hamburg Medical Chamber (Hamburg, Germany), with registration number PV6041.

### 4.3. Immunostaining

Information on antibodies and staining reagents is given in Table A1. The cells were seeded on 96-well plates (Cell+, Sarstedt, Nümbrecht, Germany) and cultivated in 50% hCSF, 45% DMEM, and 5% FBS (PAN-Biotech, Aidenbach, Germany). Then, the cultures were carefully washed with PBST (PBS, 0.05% Tween), 100 µL of formaldehyde (3.7%, Carl Roth, Karlsruhe, Germany) was added, and the cells were incubated for 15 min at room temperature (RT). After fixation, the cells were washed 3 times with PBST following a 5 min treatment with PBSX (PBS 0.1% Triton-X100, Carl Roth, Karlsruhe, Germany) at RT. The wells were blocked with PBST/5% BSA (Carl Roth, Karlsruhe, Germany) for 1 h at RT, washed with PBST, and stained with 30 µL of phalloidin-iFluor 555 conjugate (1 µL/mL PBST/1% BSA) for 1 h in the dark. After washing (3× PBST), 30 µL of primary antibody diluted in PBST/1% BSA (PBSTB) was added to each well. The plates were incubated in the dark at 8 °C overnight. The working dilution for antibodies against Axl, Sox2, Oct4, nanog, and nestin was 4 µL/mL PBSTB and 10 µL/mL PBSTB for the anti-integrin antibody. After primary antibody binding, the wells were washed 3 times, and 30 µL of a secondary anti-mouse IgG H&L antibody was added (2 µL/mL PBSTB). Incubation was carried out for one hour at RT in the dark. After washing with PBST, 30 µL of ROTI^®^Mount FluorCare DAPI (Carl Roth, Karlsruhe, Germany) was added to each well. Microscopy was carried out using an EVOS FL Auto or M7000 Imaging system (Thermo Fisher Scientific, Braunschweig, Germany). Details for antibodies and cell staining reagents used are given in Table A1.

### 4.4. Cloning of ZIKV Protein E Constructs

For the construction of E^ΔTM^gp41^TMCY^ clones, a DNA template designated pE41 was generated. Therefore, the ZIKV E^ΔTM^ fragment was introduced into pSVATG*rev* by Gibson assembly cloning, replacing the open reading frame of gp120 and the external part of gp41. Gibson assembly cloning was performed as described by Gibson [61] and using the protocol published by Samuel Miller [62]. In brief, plasmid and E DNA fragments were prepared by PCR. PCR-generated DNA fragments were added to 15 µL of Gibson-Mix (GM) to reach a final volume of 20 µL and incubated at 50 °C for 15–30 min. The GM reaction buffer was made from 64 µL of 5× ISO buffer, 0.12 µL of T5 Exonuclease (New England Biolabs (NEB), Frankfurt am Main, Germany, #M0363), 4 µL of Phusion DNA Polymerase (NEB, #M0530), 32 µL of Taq DNA Ligase (NEB, #M0208), and 140 µL of H_2_O. The Gibson 5× ISO buffer was made from 3 mL of Tris pH 7.5 (1 M), 150 µL of MgCl_2_ (2 M), 60 µL of dGTP, 60 µL of dATP, 60 µL of dTTP, 60 µL of dCTP (100 mM each), 300 µL of dithiothreitol (1 M), 1.5 g of polyethylene glycol 8000 (all from Carl Roth, Karlsruhe, Germany), 300 µL of NAD (100 mM, NEB, #B9007), and H_2_O was added to reach a final volume of 6 mL. To generate ED12 and ED3 mixed versions of ZIKV E^ΔTM^gp41^TMCY^ (pZJ, pJZ, pZU, and pUZ), DNA fragments were generated by PCR using pME-JE, pME-USU, or pME-Z1. DNA fragments were assembled by PCR and inserted into pSVATG*rev* by Gibson assembly cloning. A detailed overlook of the structures of HIV-1 gp160, ZIKV E, and its ED1, ED2, and ED3 domains, including the E41.1 and JZ construct, is shown in Figure A2.

For the construction of E1 and E2, the expression vector pME-Z1 was used as a PCR template. For the construction of E41.1 and E41.2, pE41 was used as a template. For PCR-mediated deletion, 2 ng of plasmid DNA was added to 34.5 µL of H_2_O, 10 µL of 5× Phusion HF-buffer (NEB, Frankfurt, Germany), 1 µL of forward primer (10 pmol/µL), 1 µL of reverse primer (10 pmol/µL), 1 µL of dNTP (10 mM), and 0.5 µL of Phusion-DNA-polymerase (NEB, Frankfurt, Germany). The mixture was cycled 30 times at 98 °C (45 s), 57 °C (45 s), and 72 °C (60 s/1 kb) using a Mastercycler gradient (Eppendorf AG, Hamburg, Germany). For combined ligation and template digestion, 5 µL of the PCR mixture was added to 10 µL of H_2_O, 2 µL of 10× T4 DNA-ligase buffer containing ATP (10 mM), 1 µL of T4 DNA ligase (5 U/µL, Thermo Fisher Scientific, Braunschweig, Germany), 1 µL of T4 polynucleotide kinase (10,000 U/mL, NEB, Frankfurt, Germany), and 1 µL of DpnI restriction enzyme (20,000 U/mL, NEB, Frankfurt, Germany). The mixture was cycled 50 times at 37 °C (120 s) and 15 °C (120 s), and transformation was performed using the chemically competent *E. coli* strain DH5α. Candidates were verified by DNA sequencing (LGC Genomics, Berlin, Germany).

The oligonucleotides (metabion, Planegg, Germany), used for the generation of DNA fragments are summarized in Table A2. Expression of E constructs was monitored by staining transfected COS-1 cells with an antibody broadly reactive with the flavivirus group, as shown in Figure A3.

### 4.5. Production of ZIKV-HIV Pseudotypes

Transfection of cells was carried out in 24-well format as described previously [28]. In brief, COS-1 cells were transfected with plasmids pNL*luc*AM or pNL*gfp*AM and a ZIKV-*env*-expressing plasmid, pME-Z1, -E1, -E2, -ZJ, -JZ, -ZU, -UZ, pE41.1, or -2, or with pCMV-VSV-G for comparison. The DNA/ScreenFectA (SFA, Screenfect GmbH, Eggenstein-Leopoldshafen, Germany) mixture was prepared from two solutions. Solution A was made of 30 µL of SFA dilution buffer, 9.25 µL of *env* expression plasmid (1 µg/µL), and 2 µL of pNL*luc*AM (1 µg/µL). Solution B was made of 34.5 µL of SFA dilution buffer and 30 µL of SFA transfection reagent. Solutions A and B were rapidly mixed and stored for 20 min at RT. To the DNA/SFA mixture, 312 µL of serum-free medium (Gibco™ OptiPRO™ SFM, Thermo Fisher Scientific, Braunschweig, Germany) was added, and the solution was applied to a 24-well plate with COS-1 cells at 70–80% density and incubated for 3 h at 37 °C. After transfection, the medium was exchanged for 1 mL of DMEM/10% FBS. COS-1 cell culture supernatants were harvested on day 3 after transfection, centrifuged 3 times at 10,000 rpm, and then either used immediately for infection experiments or stored for 3–7 days at 8 °C. Because part of the *env* gene in the pNL4-3 proviral plasmid was replaced with an SV40 (pNL*luc*AM) or CMV promotor/reporter cassette (pNL*gfp*AM) and the expression vectors lacked a packaging signal, the pseudotyped viruses were replication-incompetent and resulted in only a single round of infection. As an example, a schematic representation of the pseudotypes for Z1-, E1-, and E41.1-HIV*gfp* is shown in Figure A4.

### 4.6. Pseudotype Infection of Cells

U87 cells were cultivated in DMEM/10% FBS, while cells from grade 4 GBM tumors were cultivated in 50% hCSF, 45% DMEM, and 5% FBS in 96-well plates (Cell+™, Sarstedt, Nümbrecht, Germany). Infection of cells with pseudotypes was carried out as described previously [29]. Briefly, cell culture supernatants from COS-1-transfected cells were given to tumor cells at a 70–80% density. After 3 h, 100 µL of medium was added, and the cultures were incubated at 37 °C for 3 days. For luciferase detection, the cells were washed three times on day three, and 100 µL of the BrightGlow™ luciferase assay reagent (Luciferase Assay System, Promega E1483, Walldorf, Germany) was added. The plates were incubated for 5 min, and bioluminescence was measured using a 96-well plate reader (Luminometer Centro LB960, Berthold Technologies, Bad Wildbad, Germany). The green fluorescence of HIV*gfp*-infected cells was monitored using an EVOS M7000 Imaging System (Thermo Fisher Scientific, Braunschweig, Germany).

## 5. Conclusions

Improving targeted gene transfer into GBM tumor cells, especially by pseudotyping retroviral vectors with viral glycoproteins such as ZIKV-E proteins specific for certain surface markers, remains of utmost importance. Therefore, the development of such retroviral vectors is a top priority in cancer therapy, especially in the absence of specific tools that would work as vectors for therapeutic gene delivery. Therefore, any improvement can be a valuable help, and the new E variants can contribute by offering greatly improved infectivity compared to the current VSV-G. In addition, the development of such a pseudotype also has far-reaching consequences for general research in the field of flaviviruses. It is worth noting that E variants can not only be used to produce oncolytic viruses for therapy, but can also provide valuable services as a research tool to observe how the behavior of tumor or normal cells changes after the introduction of certain genes. The major advantage of the ZIKV/HIV pseudotypes is that the retroviral vectors can be easily exchanged to apply any kind of antiviral gene constructs based, for example, on CRISPR/Cas9 or siRNA technology. In addition, specific antigens can be introduced to make tumor cells more visible to the immune system. All in all, the newly presented ZIKV-E variants enable numerous in vivo and in vitro studies in the context of flavivirus research, but most importantly, they open up new therapeutic opportunities against malignant glioma cells.

## Figures and Tables

**Figure 1 ijms-24-14487-f001:**
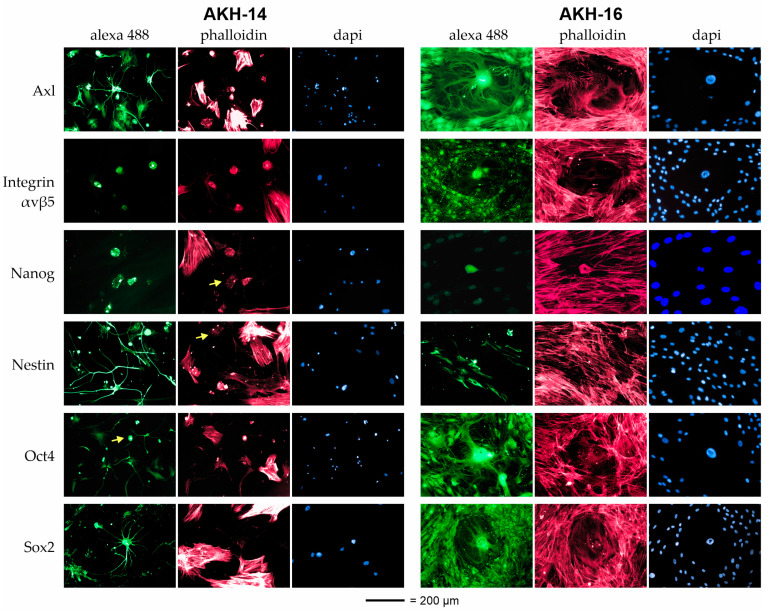
Immunofluorescent staining of representative markers nanog, nestin, Oct4, and Sox2 in cells isolated from GBM tumors. Additionally, the potential ZIKV entry receptors Axl and integrin αvβ5 were identified. For the AKH-14 and AKH-16 cell lines, staining was conducted using commercially available, marker-specific mouse monoclonal antibodies, and primary antibody detection was conducted using a secondary goat-anti mouse IgG polyclonal antibody Alexa-iFluor^®^ 488 conjugate. Yellow arrow = indication of actin-positive cells with a roundish morphology, positive for integrin αvβ5, nanog, nestin and Oct4. Alexa 488 = marker staining (green); phalloidin = actin filament (F-actin) staining (red); DAPI = nucleic acid staining (blue); scale bar = 200 µm. Details of the antibodies can be found in Table A1.

**Figure 2 ijms-24-14487-f002:**
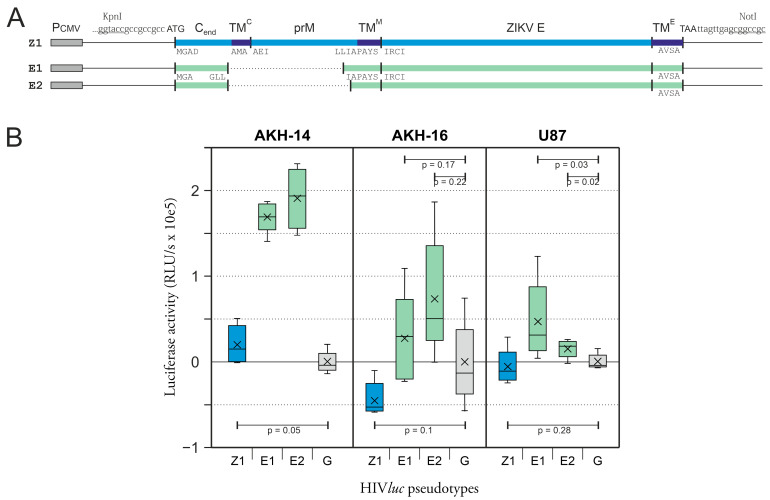
ZIKV HIV*luc* pseudotype infection of tumor cells: For infection, envelope constructs were used, all containing the native ZIKV TM^E^ domain. (**A**), Construction of envelope expression vectors; Z1 (blue), expression of ZIKV prM and E. E1 and E2 (green), expression of ZIKV E. Unlike E1, E2 lacks the amino acid Ile (I) at the C−M junction. The dotted line indicates the prM deletion. The dark blue lines indicate the TM regions of C, prM, and E. A more detailed explanation of the ZIKV prME envelope complex is shown in the appendix in Figure A2. C_end_ = c-terminal 18 amino acids of the ZIKV capsid. (**B**), Infection assay and comparative analysis of luciferase activity expressed by infected cells. Infection rates are shown relative to G-HIV*luc* (grey) as a standard of comparison. Prior to cell lysis, the respective cell culture supernatants tested negative for luciferase activity. The RLU/s data are means of fivefold infections. Z1 (blue) = prME-pseudotyped HIV*luc*; E1 and E2 (green) = pseudotyped with E; G (grey) = pseudotyped with VSV-G; underlined = recognition site for restriction enzymes; *p*-values not shown are all <0.001.

**Figure 3 ijms-24-14487-f003:**
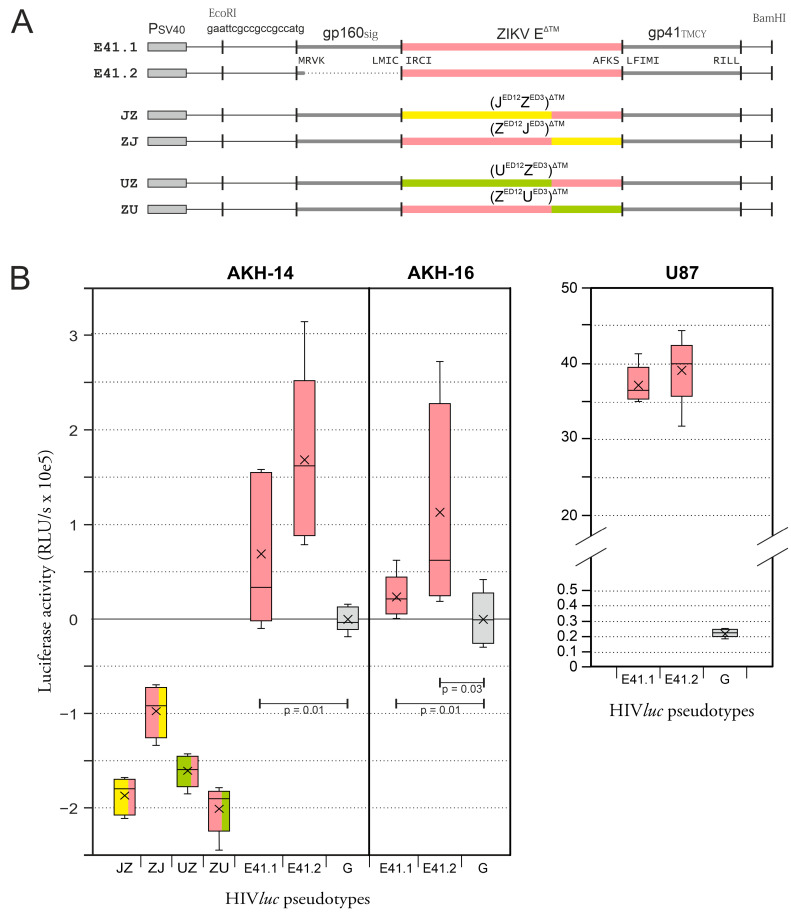
ZIKV E^∆TM^-gp41^TMCY^ HIV*luc* pseudotype infection of glioma cells: (**A**) Construction of envelope expression vectors. For E41.1, ZIKV E^∆TM^ was linked to the gp160 signal sequence and fused to the transmembrane (TM) and cytoplasmic (CY) gp41 domains. E41.2 lacks the gp160 signal sequence. Dotted line = deleted gp160 signal sequence. (**B**) U87, AKH-14, and AKH-16 cells were infected using E41.1-HIV*luc* and E41.2-HIV*luc*. Additionally, mixed ED12ED3^∆TM^-gp41^TMCY^ versions (JZ, UZ, ZJ, and ZU) and VSV-G HIV*luc* pseudotypes were used for comparison. The RLU/s data are means of fivefold infections. G = VSV envelope (VSV-G). JZ = J_ED12_Z_ED3_; UZ = U_ED12_Z_ED3_; ZJ = Z_ED12_J_ED3_; ZU = Z_ED12_U_ED3_; *p*-values not shown are all <0.001.

**Figure 4 ijms-24-14487-f004:**
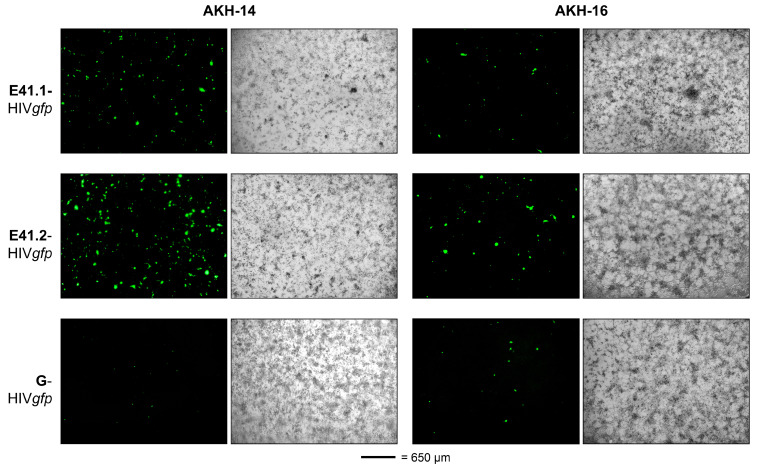
ZIKV E^∆TM^-gp41^TMCY^ HIV*gfp* pseudotype infection of tumor cells. Infection was detected by expression of the *gfp* reporter gene in single cells at 4× magnification. Green = *gfp* expression; scale bar = 650 µm. G = VSV-G.

## Data Availability

The authors confirm that the data supporting the findings of this study are all available within the figures and tables of the article.

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
