# Peer review of "Development of Zika Virus E Variants for Pseudotyping Retroviral Vectors Targeting Glioblastoma Cells"

_ijms, 2023, doi:10.3390/ijms241914487_

Round 1

Reviewer 1 Report

This topic is very interesting, but these points need to be reviewed:

- Lines 85-90: "In a first step, we examined HIV-1 viral particles pseudotyped.. from GBM tumor samples." This part is well written, but it is not clear what is the purpose of this paper. Can the authors improve this part?

- Please improve figure 1 legend.

- Lines 238-240: "It promotes cancer cell survival, proliferation, and migration by mediating interactions between cancer cells and the surrounding microenvironment" Please highlight the role of tumor microenvironment and the immune system. Consider these very recent and interesting papers: -- doi: 10.3390/neurolint15020037 -- doi: 10.3390/genes14020501

- Lines 431-432: "Unfortunately, there is a very limited number of publications to compare our... differ from previously published work." Add some references there

- A "conclusion" section is probably incorrect in this article, but a "new future strategies" section or something like this may be of interest for this paper. Please add.

Minor editing of English language required

Author Response

Reviewer 1

This topic is very interesting, but these points need to be reviewed:

- Lines 85-90: "In a first step, we examined HIV-1 viral particles pseudotyped…. from GBM tumor samples." This part is well written, but it is not clear what is the purpose of this paper. Can the authors improve this part?

Response: We have added some remarks at the end of the introduction (111-113) but also added a new chapter “conclusion” to address this point (513-529).

- Please improve figure 1 legend.

Response: The legend of Fig. 1 is now providing more details.

- Lines 238-240: "It promotes cancer cell survival, proliferation, and migration by mediating interactions between cancer cells and the surrounding microenvironment" Please highlight the role of tumor microenvironment and the immune system. Consider these very recent and interesting papers: -- doi: 10.3390/neurolint15020037 -- doi: 10.3390/genes14020501

Response: Thank you for these points. As they relate to ZIKV, we have added some comments on the microenvironment in the introduction (42-48) and in the results section (258-260).

- Lines 431-432: "Unfortunately, there is a very limited number of publications to compare our... differ from previously published work." Add some references there

Response: We have added the E/VSV-G reference at that position. For ZIKV E as an envelope for pseudotyping, that is the only one.

- A "conclusion" section is probably incorrect in this article, but a "new future strategies" section or something like this may be of interest for this paper. Please add.

Response: Thank you for this point. A new “conclusion” chapter is added to the manuscript that explains the impact on future strategies of the modified ZIKV envelope.

Reviewer 2 Report

This research made envelop constructs to create oncolytic pseutotype viruses for a future treatment of glioblastoma multiforme. It is an important research as GBM has no efficient therapy and the patients usually die within the first 6 months.

Comments/questions:

(1) Line 38. All this section is right. But you may want to cite the work of S Burzynski of the treatment of Glioblastoma with Antineoplastons.

Newly Diagnosed Glioblastoma: Partial Response and > 27 Years Overall Survival in 37-Year-Old Male Treated with Antineoplastons
(Treatment of Glioblastoma with Antineoplastons)
Burzynski Stanislaw1*, Burzynski Gregory1, Janicki Tomasz1, and Beenken Samuel2

(2) Additionally, using gene expression and a small set of genes, the overall survival of GBM was also analysis successfully identifying low risk and high risk groups.
Healthcare 2022, 10(1), 155; https://doi.org/10.3390/healthcare10010155

(3) Line 41. Could you please explain the difference between apoptosis, pyroptosis, and necroptosis?
(4) Regarding glioma stem cells. Multiple treatment strategies have been suggested at targeting GCSs, including immunotherapy, posttranscriptional regulation, modulation of the tumor microenvironment, and epigenetic modulation. You may want to include some information about this topic.
(5) Line 55. Neurotropism means infection of neural cells. But, are you refering to neurons only? GBM originates from glial cells that are non-neuronal cells in the central nervous system (brain and spinal cord) and the peripheral nervous system that do not produce electrical impulses. Do ZIKV target GSCs only?
(6) Lines 85 to 91. Could you please make a figure of the different types particles that were used? For example, the visualization may help to understand better the structure of "HIV-1 viral particles pseudotyped with ZIKV E protein" versus "prME pseudotypes" versus "pseudotypes in which the transmembrane domain (TM) of E was exchanged against 88 the TM and cytoplasmic (CY) domains of the HIV-1 glycoprotein gp41".
(7) In the infection of AKH-14 and 16 cells, did this result in increase of cell death?
(8) Are AKH and U87 cells sensitive to Temozolomide?   

Author Response

Reviewer 2

This research made envelop constructs to create oncolytic pseutotype viruses for a future treatment of glioblastoma multiforme. It is an important research as GBM has no efficient therapy and the patients usually die within the first 6 months.

Comments/questions:

(1) Line 38. All this section is right. But you may want to cite the work of S Burzynski of the treatment of Glioblastoma with Antineoplastons.
Newly Diagnosed Glioblastoma: Partial Response and > 27 Years Overall Survival in 37-Year-Old Male Treated with Antineoplastons (Treatment of Glioblastoma with Antineoplastons)
Burzynski Stanislaw1*, Burzynski Gregory1, Janicki Tomasz1, and Beenken Samuel2

Response: The aforementioned manuscript is not about GBM grade 4, which is our focus. Sorry.

(2) Additionally, using gene expression and a small set of genes, the overall survival of GBM was also analysis successfully identifying low risk and high risk groups.
Healthcare 2022, 10(1), 155; https://doi.org/10.3390/healthcare10010155

Response: The manuscript referred to is not about GBM grade 4. It is interesting, but I do not think it supports the description of the evolution of E variants in terms of producing an active pseudotype for gene transfer.

(3) Line 41. Could you please explain the difference between apoptosis, pyroptosis, and necroptosis?

Response: Thank you for that point. We have added some clarification on the differences between these effects in lines 52-54 and added two references.

(4) Regarding glioma stem cells. Multiple treatment strategies have been suggested at targeting GCSs, including immunotherapy, posttranscriptional regulation, modulation of the tumor microenvironment, and epigenetic modulation. You may want to include some information about this topic.

Response: Thank you for that point. Perhaps the first part of the introduction was scientifically clear, but a little too pessimistic about promising future strategies. We have added some information in lines 39-41 and included a reference to a review paper regarding this point.

Rong, L.; Li, N.; Zhang, Z. Emerging Therapies for Glioblastoma: Current State and Future Directions. Journal of Experimental & Clinical Cancer Research 2022 41:1 2022, 41, 1–18, doi:10.1186/S13046-022-02349-7.

(5) Line 55. Neurotropism means infection of neural cells. But, are you referring to neurons only? GBM originates from glial cells that are non-neuronal cells in the central nervous system (brain and spinal cord) and the peripheral nervous system that do not produce electrical impulses. Do ZIKV target GSCs only?

Response: Thank you for this comment. The paragraph has been reworded to make it clearer that the tropism of ZIKV is not limited to GSC. But as shown in the references, there is a high degree of permissiveness of GSC to ZIKV (62-63, 68-76).

(6) Lines 85 to 91. Could you please make a figure of the different types of particles that were used? For example, the visualization may help to understand better the structure of "HIV-1 viral particles pseudotyped with ZIKV E protein" versus "prME pseudotypes" versus "pseudotypes in which the transmembrane domain (TM) of E was exchanged against 88 the TM and cytoplasmic (CY) domains of the HIV-1 glycoprotein gp41".

Response: An explanatory graph with pseudotype images for Z1, E, and E41 was added as Figure A4.

(7) In the infection of AKH-14 and 16 cells, did this result in increase of cell death?

Response: For the development of a functional envelope protein and the measurement of infection events, such a pseudotype must not destroy cells. The viral vector itself and the envelope proteins show no lytic effect. After infection and genome integration only the reporter genes are expressed and they do no harm.

(8) Are AKH and U87 cells sensitive to Temozolomide?  

Response: We use tumor samples from primary, untreated patients. Unfortunately, we do not have information on resistance to temozolomide and cannot request patient-specific data because of the decision of our ethics committee. So, unfortunately, this question remains unanswered. However, in our opinion, it does not play such a major role with regard to the development of the envelope proteins as new tools to generate GBM-specific pseudotypes.

Reviewer 3 Report

Dear authors,

After the review process, I have several comments: why section 2 do not contain statistical data?; in the introduction, you should include new data about the microbiome influence because emerging research suggests that the gut microbiome may play a role in influencing the immune system and inflammation, which are factors involved in cancer development and progression. Changes in the gut microbiota composition may affect the body's systemic immune response, potentially impacting the development or progression of GBM; in discussions, you should add future perspectives because the microbiota can interact with the immune system in various ways. A balanced microbiome is essential for maintaining immune system health. Dysbiosis, which refers to an imbalance in the microbiota composition, could potentially affect the immune response against cancer cells, although the specific mechanisms are not fully understood.

Best regards!

Author Response

Reviewer 3

After the review process, I have several comments:

Point1

why section 2 do not contain statistical data?

Response: Data were show as a box-whisker plot, which is a common type of chart that includes five characteristic values (minimum, maximum, 1st quartile, median and 3rd quartile). Thus, we believe the most important statistical data are presented by this graph.

In addition, we have added p-values to the figures. When no p-values are provided they are below < 0.001.

The T-test for independent samples tests whether two groups differ in their mean values. It tests whether the mean values in one of the two groups are systematically higher than in the other group. However, this does not change the fact that the mean values are higher than those of the VSV-G control, which is the important message.

Point 2

in the introduction, you should include new data about the microbiome influence because emerging research suggests that the gut microbiome may play a role in influencing the immune system and inflammation, which are factors involved in cancer development and progression. Changes in the gut microbiota composition may affect the body's systemic immune response, potentially impacting the development or progression of GBM;

in discussions, you should add future perspectives because the microbiota can interact with the immune system in various ways. A balanced microbiome is essential for maintaining immune system health. Dysbiosis, which refers to an imbalance in the microbiota composition, could potentially affect the immune response against cancer cells, although the specific mechanisms are not fully understood.

                         Response: This is an interesting aspect. We think that such a topic is more relevant for an overall review that deals with the immune system and GBM in general.